# Constructive Journalistic Roles in Environments of Social Complexity and Political Crisis

Alfredo Rojas-Calderón 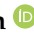

Faculty of Information Sciences, Complutense University of Madrid, 28040 Madrid, Spain; alfrojas@ucm.es

**Abstract:** This exploratory and analytical research examines secondary sources to propose a résumé of professional roles for journalists to revitalize their roles within social and political coexistence. It aims to bridge the gap between theoretical concepts and practical applications, enhancing journalists' impact on public discourse and informed decision-making. Empirical research will guide these roles' implementation in real-world journalistic practices. Twelve explanatory and constructive journalistic roles are formulated, whose relevance and application are enhanced in societies undergoing crisis situations or serious difficulties and are at risk of rupture in coexistence. In conclusion, the role of journalists in the current circumstances of crises in democratic societies requires them to not only report on events but also provide context, analysis, and solutions to the complex issues faced by society. Journalistic functions such as contextualization, public interest promotion, dialogue, and motivation are considered fundamental beyond constructive journalism or solutions journalism.

**Keywords:** journalistic roles; constructive journalism; solutions journalism; social responsibility; public problems; news

## 1. Introduction

In recent years, societies have undergone a succession of crises, yet the most common and distinct personal and social issues have either remained unchanged or become more severe. The pervasive discontent caused by troublesome circumstances is fueled by a deteriorated public atmosphere, in which the prevalence of negative news plays a significant role. According to Fitzpatrick (2022), this period is characterized by a continuous influx of adverse information. Therefore, the process of rebuilding confidence in journalism entails actively addressing the concerns of both news consumers and those who shun it due to its pessimistic nature. Serrano-Puche (2020) has confirmed that one factor contributing to the avoidance of knowledge is a lack of faith in the media.

Among the trends and predictions for this year 2023 from the Reuters Institute for the Study of Journalism on #journalism, #media, and #technology, there is projected an almost general interest among editors in explanatory journalism (94%) and in solutions' journalism (73%), as well as towards initiatives aimed at increasing the number of positive stories (48%) (Newman 2023). This responds to their concern (7 out of 10) about the trend, which the Digital News Report 2022 had already revealed, regarding the growth of evasion selective news, often related to relevant content such as politics. According to these data, since 2017, this avoidance has doubled in some countries because many people feel that news coverage is overly negative (Newman et al. 2022).

In the current context of the predominance of drama, negativity, and conflict in political news, as well as tension in public deliberation, the media and journalists are debating whether to follow this trend or reverse it by means of other informative models (Rojas-Calderón 2023b). The treatment of generalized concerns can follow the traditional negativity of journalism or, on the contrary, present a more constructive and positive approach. In that sense, Weiss (2015) has noted that professionals are exploring new methods and techniques with specific functions: how they want to bring information to

the public (disseminator), how they hold officials and companies accountable for their actions (mobilizer), how they interpret and analyze the news (interpreter/researcher), or how they incorporate the public into community debates (adversary). However, these roles are generic and very traditional, so they do not necessarily respond to the specific criteria of constructive journalism.

In consonance with Friedland and Kunelius (2023), current crisis tendencies and imaginaries are destabilizing the theory's assumptions about reality, solidarity, and personality. In their article, they examined the relevance of Habermasian theory of the "public sphere" in the 2020s, focusing on its development through the Legitimation Crisis and The Theory of Communicative Action, and suggest that these systemic disruptions are driving constitutive crises of democracy, questioning the formation of publics and highlighting the potential power of systemic powers in destabilizing late-modern lifeworlds. This can lead to a fragmented and polarized public sphere, making it difficult for different groups to engage in constructive dialogue and reach a shared understanding of reality.

Traditional media are facing declining trust, public disengagement, and revenues due to negative news. The constructive journalism movement aims to correct bias, strengthen audience–media connections, and reestablish journalism's authority. This new model encourages positive social problem-solving and offers a new option for journalists and the journalism industry (Fei 2021). According to McIntyre and Gyldensted (2018), this form of journalism requires journalists to reinterpret the principles that determine what is considered newsworthy, acknowledge that news is shaped by society, and present them in a manner that encourages positive transformation. Constructive and solutions journalism are distinct but related approaches that actually share definitions, uses, and sources (e.g., Lough and McIntyre 2021; McIntyre and Lough 2021; Aitamurto and Varma 2018). Contrary to general belief, they are rarely considered a distinct idea (for example, From and Kristensen 2018; Thier and Namkoong 2023).

Journalists can enhance the collective understanding of a community by engaging in reporting and investigative work while also providing context and explanations (Jarvis 2015). Nevertheless, a clear distinction exists between the serious subjects that journalists prioritize in news reporting, such as politics and economics, and the more popular topics that attract a wide readership, such as social issues or health (Pérez-Díaz et al. 2020). The media also should have the ability to adapt their role within a varied public sphere, providing interactive models and claiming their purpose as spaces for engagement and dialogue (Alcácer-Guirao and Fouce 2020). In this order, the constructive attitude demonstrates the accountability of both the authorities and the public in selecting suitable solutions to challenges (Aitamurto and Varma 2018).

Beckett and Deuze (2016) have warned that for journalism to maintain its value, particularly its social, political, and economic value, it must reaffirm its value of critical journalism as independent and constructive, centered on a reconceptualized idea of human interest instead of the dramatization of crises. The key is the connection between the core functions of journalism: inform, contextualize, and facilitate deliberation with audiences' affective emotions today, where positive or solution-based narratives are relevant. However, constructive journalism goes beyond simply reporting positive or good news; it involves providing in-depth analysis, offering solutions, and engaging audiences in a meaningful way. It requires journalists to move beyond the surface level and address the underlying issues that contribute to social, political, and economic challenges.

In the same way, it is not proposed that the journalist become an actor in society and politics but rather a filter in an increasingly complex society, a seeker of solutions to the problems of citizenship, and above all, an informant so that the audience can make the best political decisions (Aitamurto and Varma 2018; McIntyre and Sobel 2017; Mast et al. 2019). Constructive journalism plays a crucial role in promoting dialogue and understanding among different stakeholders, encouraging them to work together towards finding innovative solutions. It also helps to counteract the negative effects of

sensationalism and polarization in the media by providing a more balanced and nuanced perspective on complex issues.

The main objective of this work is to explore and analyze the role of journalists in the current circumstances of democratic societies' crisis and value constructive journalistic work models and solutions from this perspective. The purpose of this work is to present a proposition and résumé to highlight the evolving role of professionals in the field of journalism and their dedication to serving the public interest. By embracing these various functions, journalists are able to provide a more comprehensive and balanced perspective on important issues, fostering a more informed and engaged society. This shift towards a constructive and positive approach not only enhances the credibility of journalism but also strengthens democracy by empowering individuals to actively participate in shaping their communities.

The research has an exploratory and explanatory nature, based on qualitative research consisting of the analysis of the content from secondary sources (scientific articles, book chapters, reading basic reference books, and others) to propose professional roles for journalists that revitalize their key functions in social and political coexistence. It is, therefore, a reflective approach that will allow subsequent work to guide empirical research on explanatory and constructive journalistic roles. This approach aims to bridge the gap between theoretical concepts and practical applications in journalism by analyzing the existing literature and identifying potential areas for improvement. By proposing new professional roles, the research seeks to enhance the impact of journalists in shaping public discourse and promoting informed decision-making. The subsequent empirical research will provide a solid foundation for understanding how these roles can be effectively implemented in real-world journalistic practices.

Journalists, like other individuals, have multiple roles both inside and outside the journalistic field (Tandoc and Peters 2015). Hanitzsch and Vos (2017) reported that journalistic roles are defined and carried out on two separate levels: role orientations (which include normative and cognitive roles) and role performance (which includes practiced and narrated roles). In this way, the process model of journalistic roles posits a circular framework in which normative, cognitive, practiced, and narrated roles are interconnected through the processes of internalization, enactment, reflection, normalization, and negotiation. These processes contribute to the construction and maintenance of journalistic identities as journalists navigate their professional roles within the larger social and cultural contexts. Additionally, the model highlights the dynamic nature of journalistic roles, which can be influenced by external factors such as technological advancements and societal changes.

Mellado and Dalen (2014) state that studies have shown that role conception influences news content, but the gap between ideals and practice is inevitable, and they certainly found a significant gap between role conception and performance, particularly for service, civic, and watchdog roles. In this line, greater perceived autonomy leads to a smaller gap, while economic and political influences and belonging to a beat increase the gap. The gap varies significantly between journalists working at quality and popular presses. This study also found a large gap between journalistic role conceptions and role performance, particularly for watchdog and civic-oriented roles. The gap between roles and content is connected to personal, work-related, and media outlet characteristics. Factors explaining journalists' willingness to put their ideals into practice include economic, political, and organizational influences, as well as perceived professional influence.

Research on comparative media systems identifies distinctive models revealing key features in advanced democracies' journalistic cultures, but revisionist literature highlights limitations and hybridization of cultures elsewhere. The findings of Mellado et al. (2017) show patterns of multilayered hybridization in the performance of professional roles across and within advanced, transitional, and nondemocratic countries, with journalistic cultures displaying different types of hybridity that do not resemble either existing ideal media system typologies or conventional assumptions about political or regional clusters.

Seen in this way, journalistic roles are complex and influenced by personal beliefs, external pressures, and media outlet characteristics. They account for normative and cognitive roles, reflecting their professional identity and beliefs. However, there is often a gap between these roles, particularly watchdog and civic-oriented roles. Comparative media system research reveals different journalistic cultures across advanced, transitional, and nondemocratic countries, displaying hybridization patterns. Understanding these factors is crucial for analyzing journalists' contributions to public discourse and understanding societal issues. This means that journalists not only report on events and provide information but also analyze and interpret the significance of these events for the public itself. They go beyond just presenting facts and strive to uncover the underlying truths and implications of various situations. This in-depth analysis helps citizens make informed decisions and encourages critical thinking, ultimately leading to a more democratic society where people are actively engaged in those issues affecting them.

Western journalists are generally less supportive of any active promotion of particular values, ideas, and social change, and they adhere more to universal principles in their ethical decisions. Journalists from non-western contexts, on the other hand, tend to be more interventionist in their role perceptions and more flexible in their ethical views (Hanitzsch et al. 2011). Overgaard's (2021) study found that constructive social media posts resulted in higher levels of positive affect, self-efficacy, and perceived news credibility mediated by positive affect. This supports the broaden-and-build theory and suggests constructive journalism can help reduce news avoidance in the 21st century.

This approach recognizes the importance of engaging with audiences and their emotions, as it acknowledges that emotional responses can shape public opinion as much as drive social change. By presenting positive or solution-based narratives, journalists can contribute to a more constructive and informed public discourse. However, it is crucial to maintain journalistic integrity and avoid becoming active participants in societal and political events. Instead, journalists should focus on providing accurate information and acting as a reliable filter amidst the complexities of modern society, empowering the audience to make more well-informed political decisions.

The following controversial and divergent hypotheses are proposed in this field: Firstly, some argue that journalists should maintain a neutral stance and refrain from taking any active role in shaping society and politics. They believe that the primary responsibility of journalists is to provide unbiased information to the public, allowing them to form their own opinions and make informed decisions. On the other hand, there are those who argue that journalists have a duty to actively engage with society and politics, using their platform to advocate for positive change and address pressing issues. They believe that journalism should not be limited to mere reporting.

To review the literature and present the categories of roles of constructive journalism in crisis situations, this work followed certain specific steps:

- The relevance of the specific literature has been searched and evaluated using academic databases, online libraries, and search engines to find academic articles, books, and other sources related to the subject. The selected literature was then analyzed to identify the different categories of roles that constructive journalism plays in crisis situations.
- The literature has been analyzed by identifying the fundamental categories of roles in journalism in general and then specifically in constructive journalism, focusing on common findings related to constructive journalism, especially in crisis situations. The analysis also considered the various theoretical frameworks and methodologies employed in studying constructive journalism in crisis situations.
- The categories of roles are developed with the support of the analysis of the literature, and on the basis of the evidence that has been presented, the potential impact and effectiveness of constructive journalism in crisis situations are discussed. The potential impact and effectiveness of constructive journalism in crisis situations are discussed

by examining the various theoretical frameworks and methodologies employed in studying this field.

The transition from the literature review to the development of the roles proposed in this work involved the adoption of a systematic approach to ensure rigor and transparency in the process. To this end, the gap in current research was first identified, starting with a review of the existing literature, synthesizing key findings, and identifying areas in which further research and development are justified. This laid the groundwork for the proposal's formulation in terms of its relevance and the need for advancement in the field.

The literature review's knowledge informed the development of the roles presented. This study's contribution to the field's literature is based on articulating a proposal complementing and advancing knowledge, questioning dominant assumptions, and extending existing theories for further research. By contextualizing the work within the broader academic conversation and demonstrating its importance, it has sought to emphasize the novelty and relevance of this contribution.

In addition to the methodological rigor, the iterative nature of the proposal–development process was paramount. Emphasis has been placed on the continuous refinement and validation of proposed roles through feedback loops. This iterative approach allowed us to fine-tune these proposals, address any potential limitations, and ensure that they are robust and well-founded. Furthermore, the practical implications of the roles presented have been insisted upon. By highlighting the practical relevance of this proposal, the aim has been to bridge the gap between academic research and practical application, ultimately increasing the impact of this study on addressing real-world challenges.

## 2. Literature and Analysis

Hanitzsch and Vos (2018) state that journalism scholarship often overlooks non-democratic and non-Western contexts and forms of journalism beyond political news. In line with these authors, the journalistic roles are constructed through discursive institutionalism, shaping journalism's identity and societal position. They play crucial roles in political life, addressing six essential needs, and in everyday life, affecting consumption, identity, and emotion influenced by institutional norms and practices. Journalists play various roles, including analytical–deliberative, critical–monitorial, advocative–radical, developmental–educative, and collaborative–facilitative. They advocate for political change, raise public awareness, and support the government in achieving development and social well-being. They act as the 'Fourth Estate', holding powers accountable and promoting social change beyond the discursive realm of journalism.

In a previous study, Hanitzsch et al. (2016) investigated the professional role orientations of journalists, focusing on three aspects: setting the political agenda, influencing public opinion, and advocating for social change. The research found that journalists are more willing to intervene in society when they work in public media organizations and in countries with restricted political freedom. Political freedom plays a major role in this relationship, with journalists in politically less free countries being more likely to embrace an interventionist role in society. Journalists' professional role orientations are also rooted in perceptions of cultural and social values. Journalists were more likely to embrace an interventionist role when they were more strongly motivated by the value types of power, achievement, and tradition.

This study also found evidence of a relationship between journalists' professional role orientations and cultural values, such as power, achievement, and tradition. Power values are more directed towards the ultimate goal of social change than to the discursive mechanisms to achieve this goal. Achievement values target the political discursive realm, making journalists more likely to set the political agenda and influence public opinion (Hanitzsch et al. 2016). However, as Aitamurto and Varma (2018) explain, solutions journalism and constructive journalism do not recognize the role of journalism in setting the agenda: the solutions discussed in the news are more easily legitimized and normalized in public discourse.

Another extensive work by Hanitzsch et al. (2011) examined the perceptions of journalists in various countries and transitional democracies and revealed that political factors, such as political climates and media laws, significantly influence journalists' ethical views. Specifically, journalists in countries with hostile political climates exhibit smaller power distance and need more flexibility in responding to ethical dilemmas. Under legal uncertainty and weak jurisdiction, journalists need more flexibility in responding to ethical dilemmas and focusing more on the potential consequences of their decisions. Thus, non-western journalists tend to approve of contextual and situational ethical decision-making and the application of individual standards more than their Western counterparts.

Based on another global survey, Hanitzsch et al. (2010) identified six distinct domains: political, economic, professional, procedural, and organizational influences. In this case, political influences include government officials, politicians, and censorship, while business people, such as entrepreneurs and industrialists, are also considered. Economic influences, on the other hand, involve factors that directly affect news organizations. The moderate importance of political and economic factors may contradict intuition, as they are rarely experienced directly by the average journalist. The power of these influences might be absorbed by news organizations and subsequently filtered, negotiated, and redistributed to individual journalists.

A similar study by Hanitzsch and Mellado (2011) confirms that political and economic factors are the most important denominators of cross-national differences in journalists' perceptions of influences. In fact, perceived political influences are related to objective indicators of political freedom and ownership structures across the investigated countries. Economic influences seem to have a stronger impact in private and state-owned media than in public newsrooms, but they are not related to a country's economic freedom. This work also found that journalists working in private and public news media also differ with respect to their perceived level of political influence. Journalists in private newsrooms reported significantly more political pressure, albeit the difference was less striking. These differences may be partly related to characteristics of national media systems, as the interests of political and economic elites are often so interrelated that it may be difficult for journalists in state-owned media to distinguish between genuinely political and economic influences.

The findings support the widely assumed supremacy of political and economic factors as the driving forces behind differences between media systems and journalistic cultures. Perceived political influences are clearly related to objective indicators of political freedom and ownership structures across the investigated countries. However, there may exist additional cultural factors and organizational characteristics that shape the perceptions that journalists have regarding economic influences. These cultural factors could include societal norms, values, and historical contexts that influence how journalists perceive and interpret economic influences. Furthermore, organizational characteristics such as media ownership concentration and editorial policies may also play a role in shaping journalists' perceptions of economic influences on media systems.

The findings of Aitamurto and Varma (2018) show that metajournalistic discourse indicates tension over the normative roles of journalism. They explain that the metadiscourse surrounding these two types of journalism reveals a contradictory approach to journalism's role in collective action: on the one hand, they prohibit journalism from mobilizing social change by publishing calls for action or recommending solutions; on the other hand, they maintain journalistic ideals of objectivity and accuracy by creating a more comprehensive and representative picture of the world. This tension between normative roles of journalism is often seen in the debate surrounding advocacy journalism and objective journalism. Advocacy journalism, characterized by its explicit support for a particular cause or viewpoint, is criticized for blurring the line between reporting and activism. On the other hand, objective journalism, which strives to present information without bias or personal opinion, is sometimes accused of being detached from the realities and needs of society. Despite these conflicting perspectives, both types of journalism contribute to a more nuanced understanding of complex issues.

The emphasis on traditional values in journalism may also contribute to a resistance to embracing new and innovative approaches to addressing societal issues. Therefore, constructive and solution-oriented journalism models need to define their own roles. Thus, it is pertinent to analyze the studies already carried out while at the same time identifying and characterizing other types of roles that serve as hypotheses for new applied research. This approach would not necessitate journalists abandoning their negative biases in the news but rather expanding their perspective. Rather than restricting their coverage to those in positions of authority, they should highlight constructive dialogues that are frequently obscured in media. By doing so, journalists can provide a more balanced and comprehensive view of the issues at hand, fostering a better understanding among the public. Additionally, this approach can help shift the focus from sensationalism and conflict to solutions and progress, ultimately contributing to a more informed and engaged society.

The metadiscourse analysis shows that both types of journalism justify their existence by the need to solve vexing social issues and by the increasing number of social innovations to address these problems. However, they distance themselves from being proponents of social good by claiming to only objectively cover solutions without preferences or values affecting the selection of those solutions. They also avoid acknowledging journalism's role in agenda-setting, as solutions covered in the news are more easily legitimized and normalized in public discourse. Thus, metajournalistic discourse suggests a new role for journalism as a "change-agent" rather than a detached observer, aiming to provide solutions and constructive journalism (Aitamurto and Varma 2018). However, these findings should be interpreted with caution as they are based on a limited sample size that comes from within the same organizations and may not be representative of all journalists' perspectives, neither external nor independent voices.

Smeenk et al. (2023) used the ethos perspective, which holds that the strategic self-image of a journalist is crucial for the performative potential of journalism, even in detached "objective" journalism. An ethos explains how journalists build on and rework epistemological frameworks to ensure the text's performativity. It offers an integrated framework for studying relationships between news texts, news production, contexts, and audiences, highlighting how values such as reliability, authenticity, or objectivity are projected, circulated, and attributed in the journalistic field and the information ecology. The image that the journalist creates of themselves is intricately connected to existing conceptions of journalism. In order for a news story to fully realize its performative potential, the journalist's ethos must align with the established conventions of the sector.

A recent article presents a comparative assessment of normative journalistic roles based on qualitative responses from journalists in 67 countries and found that journalists see their normative roles primarily in the political arena. In non-Western countries, journalists advocate for intervention in social processes and a constructive attitude towards ruling powers. The normative core of journalism is built on the news media's contribution to political processes and conversations, while other areas, such as self-management and everyday life, remain marginalized (Standaert et al. 2019). This focus on the political arena is driven by the belief that journalism plays a crucial role in holding those in power accountable and ensuring transparency in governance. However, it is important to note that this emphasis on politics may limit the coverage of other important aspects of society, such as social issues and everyday concerns of the general public. Therefore, there is a need for a more balanced approach that encompasses a wider range of topics to truly serve the public interest.

Based on a survey among political journalists in Denmark, Germany, the UK, and Spain, Dalen et al. (2012) revealed that role conceptions vary more across countries than within them. Spanish journalists perceived their role as sacerdotal and partisan, while British journalists were more entertainment-oriented. The unique Spanish perspectives on roles were evident in the reporting style of Spanish newspapers, which prominently featured political news, presented with less emphasis on conflict or competition, and displayed a biased tone towards politicians. These findings suggest that cultural factors

play a significant role in shaping journalists' perceptions of their role and subsequently influencing their reporting style. It is possible that the sacerdotal and partisan role conceptions held by Spanish journalists reflect the historical and cultural context of Spain, where the influence of religion and political ideologies may have become more pronounced. This highlights the importance of considering cultural differences when analyzing media landscapes and understanding how they impact journalistic practices.

In Germany, constructive journalists have eight role dimensions, including the social integrator, transformation agent, Active Watchdog, Emotional Storyteller, and Innovation Reporter. Krüger et al. (2022) found that these journalists aim to control political and business elites, motivate participation, and contribute to social change. They are not only solution-oriented but also work normatively, politicized, and attached to specific issues, and may also be watchdogs of political and business elites in a Western tradition. These role dimensions reflect the diverse responsibilities and goals of journalists in contemporary society. By acting as social integrators, they strive to bridge gaps and foster cohesion within communities. Additionally, their role as transformation agents highlights their commitment to driving positive change and challenging existing power structures.

A study of 21 Egyptian and Tunisian journalists for Allam (2019) reveals the importance of constructive journalism during transitional periods. It focuses on regaining audience confidence, fighting terrorism, serving the public interest, and reviving the mainstream media's economy. However, challenges include political power structures, private ownership, and the possible misinterpretation of the term constructive. The study proposes an integrated strategy between mainstream media and social media platforms, based on "constructive-interactive", to reconcile media entities and audiences. Allam (2019) argues that mainstream media should engage with audience interaction on social media, reflect on government policies, express concerns, and encourage problem-solving.

Another research by Ojala and Pöyhtäri (2018) examined how Finnish journalists conceived their professional roles during the 2015–2016 refugee crisis, adopting a social-interactionist approach and analyzing open-ended interviews with 24 journalists. This study reveals how political context, interactions with key reference groups, and the political context shape journalists' conceptions of their tasks and duties. The study highlights the tensions involved in journalistic balancing and negotiation between various role conceptions, especially in a Europe marked by multiculturalism and anti-immigrant movements. The social-interactionist approach emphasizes the relational nature of journalistic roles to reference groups and social contexts.

A survey of US newspaper journalists revealed that they highly value professional roles associated with contextual reporting, including the 'Contextualist' who places high value on being socially responsible and accurately portraying the world. Journalists' belief in activist values, such as setting the political agenda and pointing at possible solutions, predicted more favorable views of all three forms of contextual journalism, while belief in an adversarial attitude predicted less favorable views of restorative narrative. The most prized function was the hybrid, which combines interpretive roles with 'just the facts' disseminator roles. Job function heavily influences attitudes toward this reporting style, with the advocate/entertainer function being the strongest overall predictor of positive attitudes toward contextual journalism genres (McIntyre et al. 2018; Abdenour et al. 2018).

These findings, which involve input from both journalists' and citizens' perceptions, have been tested by these authors in other more recent works. Dahmen et al. (2019) found that journalists' role functions, which include their identification with key journalistic values, significantly predict the coverage of victims and survivors of mass shootings. Contextualist journalists, who contribute to society's well-being and accurately portray the world, strongly believe mass shooting coverage is an ethical issue. Interpretive/disseminator journalists, who are objective, have a favorable attitude towards covering perpetrators. Advocates and entertainers who set the political agenda and advocate for solutions are more supportive of victim coverage. The research surveyed US journalists on mass shooting coverage and its improvement. The majority agreed that the media does a good job but also

supported longer coverage and a societal focus on community resilience. They believed that coverage could be improved by including potential solutions.

In this same line of work, Abdenour et al. (2021) reaffirmed that the public highly valued contextual journalism roles, with five of the eight highest-rated roles reflecting contextual principles. The contextualist function was more important to citizens than any of the four traditional functions (disseminator, interpretive, populist mobilizer, adversarial). This suggests that contextual reporting is important to audiences and could be key to strengthening the news media's public stature. However, audiences had mixed feelings about other journalistic functions, with some rating disseminator roles high and others low. Interpretive role valuations were slightly above average, and the individual role of investigating government claims was important to citizens. Audiences gave the lowest ratings to adversarial roles but were more accepting of the journalist-as-adversary compared to news workers.

These authors have highlighted the importance of contextual journalism and the influence of journalists' roles and values on their coverage of various issues. These studies emphasize the role of journalism in setting the political agenda and advocating for solutions. The research also found that journalists' identification with key journalistic values can significantly influence coverage of events like mass shootings. The study by Dahmen et al. (2019) showed that contextualist journalists were supportive of covering victims and survivors of mass shootings, while interpretive/disseminator journalists focused on perpetrators. Abdenour et al. (2021) reaffirmed the public's high appraisal of contextual journalism roles, with contextualist functions being rated higher than traditional roles. These studies provide valuable insights into the role of journalism in shaping public discourse and understanding and the importance of contextual journalism in strengthening the media's relationship with its audience.

On the other hand, Thier and Namkoong (2023) examine the differences between solutions journalism and constructive journalism, focusing on their components and their impact on journalistic practice. Solutions-oriented journalism focuses on covering credible responses to social problems to enhance journalists' social responsibility. It relies on frames and journalistic roles to explain and contextualize issues, sometimes creating unique frames or roles. These frames and roles are typically normative. Constructive and solutions journalism mirror normative journalistic tensions between active and passive roles in journalism and advocacy, although these tensions are more acute and less normative in developing national contexts. The authors argue that normative journalistic tensions between active and passive roles in advocacy and journalism are mirrored in constructive and solutions journalism but are more severe in developing national contexts.

In addition, from the perspective of Mäder and Rinsdorf (2023), context is crucial in news reporting as it enables the audience to better grasp the significance of societal issues by carefully selecting and presenting the relevant facts. But they do not want to go far beyond a "just the facts approach", in contrast to McIntyre et al. (2018). These authors' argument posits that constructive journalism revitalizes traditional journalistic values by presenting societal issues with relevant context, thereby enabling informed discourse. They contend that certain journalistic strategies may have become obsolete due to their potentially counterproductive effects in the current media landscape. This study concludes that constructive journalism encourages journalists to evaluate arguments' merits, allowing readers to assess situations while not limiting reporting to power figures but highlighting productive discourse in digital media.

Li (2023) examines solutions journalism by content characteristics rather than perceptions. She found that 70% to 90% of solutions-driven articles met nine out of eleven attributes of solutions journalism. However, most articles did not mobilize audiences to participate in the solution process. The study also explored the patterns of responses and respondents in the pandemic coverage, identifying differences across countries. Solutions journalists focused on the government's containment of the virus, followed by civic societies and citizen groups' coping and adapting. The approach found that solutions journalism

demonstrated interventionist, facilitator, and civic-oriented roles in their coverage but underplayed both the service role and watchdog role. The U.S. coverage demonstrated fewer interventionist and facilitator roles but a more civic-oriented role.

The significance of providing context in news reporting to help audiences better grasp societal concerns is highlighted by Thier and Namkoong (2023) and Mäder and Rinsdorf (2023). They differentiate between two types of journalism: solutions journalism and constructive journalism. The latter aims to restore the credibility of journalism by providing a more nuanced and accurate presentation of social concerns. Both studies recognize the challenges of maintaining a balance between active and passive roles in advocacy and journalism, as well as the possibility that some journalistic tactics may have become antiquated. According to Li (2023), analysis of solutions journalism publications shows how, although focusing on contents and attributes that match most criteria from a solution-oriented approach, those prove to be unable to inspire readers to take action.

Another article that explored the role of mass media in providing accurate information during crises suggested that journalism is as much about ritual and meaning-making as about providing information, particularly through live, on-the-spot journalism. The study revealed that key journalistic strategies, such as immediacy and competition, are motivated by rituals related to affirming community and journalistic organizational needs as by informational motivations. Thus, journalists should consider the roles of psychologist, comforter, and co-mourner in times of crisis, especially in a live, 24 h news culture (Riegert and Olsson 2007). These roles are essential in order to fulfill the community's need for reassurance and support during difficult times. By understanding the psychological and emotional impact of crises, journalists can effectively engage with their audience and provide a sense of unity and empathy. This approach not only strengthens the bond between journalists and their community but also enhances the overall quality of news reporting.

Kibarabara (2023) considers of high importance the fact that journalists conceive their roles along the dimensions of the critical–monitorial or of interventionism, as it presents insight regarding the tensions between the role conceptions at the individual journalist level and the role expectations at the institutional level and how these might inform the practicality and achievability of constructive journalism in their daily practice. By understanding the tensions between individual journalist role conceptions and institutional role expectations, journalists can navigate the complexities of their profession more effectively. This knowledge allows them to assess the feasibility of implementing constructive journalism in their daily practice and make informed decisions about their role as critical monitors or interventionists.

Solution journalism requires unique skills and resources compared to traditional reporting. Journalists must identify and research stories offering solutions, tell them compellingly, and avoid oversimplifying complex issues. It can change news consumption and inspire action at the same time. The practice depends on newsrooms' reporting styles, but solution journalism and constructive journalism are likely to be preferred in the future to meet reader information needs and create a society (Thanh et al. 2023). By focusing on solutions, journalists can provide a more balanced and hopeful perspective on the world, which can help combat the negativity bias often associated with traditional news reporting. Additionally, solution journalism has the potential to foster collaboration and dialogue among different stakeholders, leading to more effective problem-solving and societal progress.

On the other hand, Standaert et al. (2019) explored the roles journalists play when covering social justice topics and how these roles reveal emotions and self-expression values in news production. The work finds that journalists aim to guide, motivate, and inspire audiences by using emotion in their stories. Journalists negotiate between being rational and having a social impact while keeping emotions and a desire for social change at arm's length. Tandoc and Peters (2015) proposed the concept of dual roles. They concluded that some coordinators prioritize their role as journalists, while others work on high-profile cases, focusing on their role as coordinators. The longer a coordinator serves, the more

familiar they are with the demands of their role, reducing internal conflict. It explains that individual differences may also influence managing multiple responsibilities.

Research on multiple roles should treat the experience of dual roles as a continuum influenced by situational and personal factors. Role integration can help cope with role conflict, but it can be impeded by the journalistic norm of autonomy. This study expands our understanding of the collaborative role of the news media, as it coexists with a monitorial role on a routine basis. An informed understanding of journalists and their actions considers that they constantly balance multiple social and occupational roles (Tandoc and Peters 2015).

In one of their last works, Mellado et al. (2020) investigated the gap between individual role conceptions and the average role performance of journalists in nine countries from Latin America, Western Europe, and Asia. The research focused on institutional influences on the conception–performance gap at three levels: individual, organizational, and societal. The results show that the gaps are largest for the civic and watchdog roles, which are most connected with the public functions of journalism. The study also found that the size of the gaps differed more clearly between journalists and media organizations than between countries. Institutional predictors analyzed for this study included ownership, codified editorial policies, and media audience orientation.

The gap between journalists' ideals and media organizations also depends on whether newspapers address diverse or politically interested audiences. Elite newspapers and their journalists appear less inclined to pursue a consumer-oriented service role, resulting in smaller conception–performance gaps. The infotainment gap is not affected by the media's audience orientation, likely due to increasing commercial orientations. The findings show that the infotainment gap was larger in newsrooms with established editorial policies, while the interventionist gap was smaller in the same newsrooms, so it clearly suggests a persistent prevalence of the disseminator role as a traditional journalistic standard codified in today's media organizations through editorial policies, normalizing an essential journalistic ideal (Mellado et al. 2020).

The greater influences of codified editorial policies on the infotainment gap may also be a sign of a still-prevalent orientation of newspapers to more classic hard news coverage compared to the focus on sensationalistic soft news portrayals. On the other hand, this study reveals that State-owned newspapers have strong gaps in the conception–performance gap for interventionist and loyal–facilitator roles, with journalists' ideals less reflected in news coverage. This is consistent with earlier research showing that State-owned media exert more external control over journalists, favoring a more state- or government-aligned coverage (Mellado et al. 2020).

Finally, individual journalistic autonomy plays a role in explaining the conception-performance gap in public service-oriented roles. Journalists who perceive more journalistic autonomy assign more relevance to both roles without necessarily being better reflected in their organizations' actual news coverage. The causal direction of the relationship between perceived autonomy and role conceptions cannot be addressed by the study, but future research should investigate whether journalists pursuing watchdog and civic roles are more autonomous or if more autonomy gives an impetus for journalists to reflect more on the importance of these standards (Mellado et al. 2020). According to McIntyre et al. (2023), the lack of cohesion between the idealized conception and performance of the Watchdog role suggests that institutional and social system factors can outweigh individual predispositions.

In another article, Hermida and Mellado (2020) present a conceptual framework for analyzing journalistic norms and practices on social media platforms, specifically Twitter and Instagram. It proposes five analytical dimensions: structure and design, aesthetics, genre conventions, rhetorical practices, and interaction mechanisms and intentionality. The study acknowledges that social media is not homogeneous and that different platforms embody their own internal ideals. Further research could explore how genre conventions may impact journalistic roles, such as the role of the dispassionate reporter on Twitter and

social influencer on Instagram. The analytical dimensions aim to advance research into the reinterpretations and redefinitions of journalist practices outside of institutional news media spaces.

The research examines how the sociopolitical backdrop and the positions within journalism impact the views of professional responsibilities in different journalistic cultures. It also analyzes how these roles are reflected in the substance of news. The research is also conducted from a cross-cultural analysis standpoint. Surveys frequently express roles as archetypal forms. Constructive journalism is regarded as a crucial response to the dominance of negative news, including war. From a journalistic standpoint, proponents of explanatory and constructive models advocate for a different prioritization of news values in the decision-making process, giving preference to societal advancement, problem-solving, and a focus on the future (Hermida and Mellado 2020).

These studies offer valuable insights into the relationship between journalists' role conception and media organizations, particularly in the context of public service-oriented roles. However, there are several points to consider in terms of a critical analysis of these findings. Overall, these studies offer valuable insights into the complex interplay between journalists' role conceptions and media organizations, as well as the potential impact of external factors such as audience orientation, editorial policies, and social media platforms. But, given the circumstances of the political and institutional crises that many Western countries are going through, it is very important to know and understand how journalistic roles are defined and performed in different contexts. For this reason, it is also pertinent to introduce the foundations of constructive journalism and solutions journalism from this perspective. It certainly is about exploring the societal dimension of journalistic roles.

This is especially true given the increasing focus on the societal impact of journalism and the role that media plays in shaping public perceptions and discourse. Constructive journalism and solutions journalism offer promising approaches to addressing this societal dimension of journalistic roles. By introducing and exploring these journalistic approaches, it is possible to gain a deeper understanding of how media can contribute to positive societal change and address the challenges posed by political and institutional crises. It is paramount to consider how these approaches can be applied in different cultural and political contexts, taking into account local factors and conditions. Moreover, it is crucial to examine the role of journalists as agents of change within their communities. By adopting constructive and solution-oriented approaches, journalists can empower citizens to engage in civic dialogue and collective action, ultimately contributing to a more informed and active citizenry.

## 3. Proposition

A search of the relevant literature was conducted using academic databases and other sources. The common categories have been extracted and synthesized, which has allowed the development of new ones regarding journalistic roles in the terms proposed in this investigation. This means closing the gap between theoretical aspects and application, for which it will be necessary to validate and refine the new categories in future research and practices.

In order to implement solutions journalism, McIntyre and Lough (2021) determined that the problem-solving process should be the focal point of the narrative. The story should prioritize providing additional details about the response rather than focusing solely on the problem. Specifically, it should elaborate on the implementation process and provide indicators that highlight the impact and limitations of the response. Similarly, the solution must be concrete rather than hypothetical, and the narrative must be rigorous and comprehensive. Additionally, it ought to provide details on mobilization, particularly regarding how to actively contribute to social transformation.

Casares (2021) asserts that the construction model is characterized by its emphasis on the achieved outcomes, which are supported by data. It also involves examining the constraints of projects, quantifying their social influence, and extracting insights from past

experiences. Its primary objective is to facilitate the connection and collaboration between individuals and organizations involved in addressing a particular problem. It aims to foster inclusivity by incorporating diverse perspectives into the discourse and establishing platforms for engagement.

The primary objective of constructive journalism is to effectively organize and present the intricate aspects of social reality by carefully choosing and arranging theme frameworks. This sort of journalism strives to offer a thorough comprehension of many societal matters by meticulously arranging and presenting material in a logical manner. It assists readers in comprehending the complex network of society interactions, providing essential perspectives and examination. Moreover, explanatory journalism endeavors to narrow the gap between intricate topics and the general populace, guaranteeing that essential information is easily understandable and comprehensible for everyone.

Constructive journalism empowers readers by simplifying intricate concepts and employing straightforward language, enabling them to comprehend and connect with significant subjects that could otherwise appear daunting or perplexing. This sort of journalism not only educates the general audience but also promotes analytical reasoning and well-informed decision-making, cultivating a more involved and participating society.

The conventional indicator function, which primarily highlights public concerns, needs to be broadened to encompass not only the analysis of these societal issues but also, and more importantly, the identification of their remedies, the allocation of accountabilities, and the actions that individuals and their communities can take to address and confront them. The utilization of this extended indicator function enables people and groups to have their own ability to take proactive measures and create a beneficial influence on the prevailing difficulties. By offering comprehension of the issues as well as effective remedies and opportunities for transformation, it fosters active involvement and contribution towards the establishment of an improved society for everyone.

Given the prevalence of conflict frames, the solutions journalist serves as a mediator between different parties or between the entirety and its components. This type of journalist has the ability to actively engage in conflicts rather than only observing from the outside, therefore contributing to the development of solutions alongside the key individuals involved. The active engagement of the solutions journalist not only facilitates comprehension of the intricacies of the issue but also promotes empathy and cooperation among the parties involved. Through proactive interaction with the main characters, the solutions journalist might discover novel concepts and tactics that could have been disregarded, eventually resulting in more efficient and enduring resolutions to conflict scenes.

The job of indicator includes evaluating other options that may be more successful in comparable difficult situations. Additionally, it assumes the role of a facilitator by possessing knowledge of potential solutions to a broad issue and creating opportunities for debate and learning among the individuals and groups concerned. The constructive or solutions journalist serves as a civic educator, advocating for the significance of community, collaboration, and social harmony. The constructive or solutions journalist fosters active participation in problem-solving processes by emphasizing alternatives and facilitating discourse among individuals and groups. This strategy not only promotes a feeling of empowerment but also contributes to the development of more robust and resilient societies. These journalists, by means of their reporting, motivate collective action and inspire citizens to assume responsibility for fostering constructive transformation within their communities.

The traditional function of a "watchdog" is redirected towards the essential monitoring of the organizations responsible for handling or overseeing the resolution of social issues. Explanatory journalism necessitates people who possess talents associated with the societal aspect of journalism, including the capacity to inspire action on matters that concern citizens. These experts not only educate the public about social issues but also offer comprehensive analysis and context to facilitate individuals' comprehension of the underlying causes and possible remedies. Explanatory journalists strengthen communities by emphasizing

the significance of citizen involvement and activity, enabling them to hold institutions responsible and collaborate towards a more promising future.

In this sense, the following explanatory and constructive journalistic roles are formulated, whose relevance and application are enhanced in societies undergoing crisis situations or difficulties and at risk of rupture in coexistence:

1. Structuring: The constructive journalist performs the conventional duties of choosing and organizing topics of widespread importance and their presentation but does so by creating thematic frameworks that depict and streamline the intricate nature of social reality. In order to do this, it employs not only the elucidation of the issue scene but also the exposition of its means for solutions. This methodology facilitates readers' comprehension of the interdependence among many elements and motivates them to adopt a more comprehensive perspective of the world. The constructive journalist enhances readers' comprehension of the root causes and potential remedies for difficult circumstances, enabling them to actively engage in collaborative efforts towards a more promising future. Moreover, this role of the structure serves to mitigate the problem of information overload by arranging intricate subjects in a manner that is readily comprehensible to the public.

2. Contextualizer: This journalistic position involves providing a comprehensive understanding of the situation by placing it within a wider social and institutional framework. The text discusses the sources, effects, and remedies of the problem, including a comparison with similar cases. This position facilitates the public's comprehension of the problem's gravity and its broader ramifications for society. Journalists facilitate readers' understanding of the issue by offering context, allowing them to consider many viewpoints and make well-informed choices on possible resolutions. Furthermore, providing context to the situation enables readers to see and pay attention to patterns and trends, resulting in a more profound comprehension of its underlying causes and possibly long-lasting consequences. It also helps explore both the difficult situations and the related responses, offering a thorough investigation and supporting information on the subject.

3. Promotor of public interest: Explanatory and solution journalism should provide valuable and practical knowledge. In order to accomplish this, it must strive to shed light on the discourse around matters concerning or impacting both the majority and minority groups within society. Furthermore, journalism serves to uphold the liberties of individuals and collective rights, safeguarding the welfare of the entire populace. Public service journalism may establish a platform for disadvantaged voices to be heard and recognized by promoting inclusion and diversity. The role of accountability in holding authority responsible and maintaining openness in decision-making processes is essential since it ultimately enhances democracy and fosters a fairer society. Moreover, public service journalism has the potential to empower individuals by equipping them with the essential knowledge and resources to effectively engage in creating their communities and making well-informed choices.

4. Facilitator of social integration: The role dimension in constructive journalism emphasizes the integration of diverse social views and the promotion of social cohesion. Social integrator journalism fosters inclusivity and harmony in society by emphasizing multiple perspectives and fostering mutual understanding among different groups. The objective of this aspect of constructive journalism is to cultivate a feeling of belonging and mutual respect, thus promoting a more inclusive and peaceful community. In addition, through the prioritization of mutual objectives and collective principles, social integrator journalism has the potential to enhance the general welfare and the consolidation of social cohesion.

5. Motivator or inspirer: At this stage, the argumentative element concerning the proposed solutions is combined with the emotional element, which addresses the reasons for taking action in response to feelings of helplessness, impotence, apathy, and societal pessimism. The primary objective is to promote both individual initiative and

collaborative efforts in seeking solutions while also stressing the importance of social learning as a shared objective within the community. Although they may or may not identify the responsible parties for implementing the solutions, they usually provide the desire to take action. In this case, the function of narrative is to communicate solutions and motivate transformation.

6.  The transformation agent is a key component of constructive journalism seeking to facilitate good change and social development. Constructive journalism's transformation agents actively strive to foster positive change and societal transformation by emphasizing solutions and motivating action. They not only report on issues but also highlight people, organizations, and projects that are effecting positive change. By means of their narrative, their objective is to inspire and encourage people to actively participate and have a positive impact on the world. Transformation agents play a vital role in developing a more positive and proactive society by magnifying success stories and supporting social innovation.

7.  As promoters of bridges, explanatory and constructive journalism involve facilitating expression and fostering understanding between diverse socioeconomic and political groups, with the aim of fostering tolerance and collaboration, as well as creating a more inclusive society. This role requires journalists to proactively pursue diverse points of view and amplify the voices of disadvantaged populations, promoting coexistence and dismantling obstacles. Bridge builders serve as intermediaries, facilitating connections between antagonistic groups and promoting common sense and shared goals between people with divergent perspectives. It is, therefore, about promoting discourse and intercomprehension between disparate sectors or rival factions.

8.  The community advocate job involves journalists proactively interacting with local communities to comprehend their needs and problems and subsequently reporting on matters that directly impact them, therefore amplifying the voices of underrepresented groups. This function surpasses conventional reporting by aggressively pursuing narratives and perspectives that may be overlooked by the mainstream media. Journalists may effectively raise awareness of the challenges faced by marginalized populations and actively promote efforts for reform by amplifying their voices. In addition, community advocates may promote discussion and cooperation among community members and decision-makers, guaranteeing that their problems are acknowledged and resolved. The objective is to promote resilience and optimism by showcasing narratives of individuals overcoming challenges, creating community collaboration, and facilitating constructive transformation in the midst of hardship.

9.  The role of the social integrator of journalists involves actively promoting inclusion, dismantling barriers, and reducing tensions that exist between various groups in society. This may be accomplished by using strategies such as coordinating communal gatherings, fostering inclusivity within the media portrayal, and championing equitable prospects for individuals irrespective of their backgrounds, origins, and preferences. Through proactive efforts in promoting social integration, journalists and community activists may play a significant role in fostering a more unified and peaceful society.

10.  Diffuser of innovative solutions: The reporter's function entails the duty of journalists to emphasize and exhibit inventive responses to social problems. This position involves actively searching for and documenting innovative concepts, technologies, and methodologies with the capacity to generate beneficial transformations. Through the act of emphasizing these groundbreaking ideas, journalists have the power to motivate and promote the implementation of novel responses that can successfully tackle societal concerns. Moreover, this position also entails the responsibility of holding those in positions of authority responsible for executing these solutions and guaranteeing their availability to all segments of the population. Thus, they serve as a source of inspiration, encouraging others to think innovatively and discover fresh answers.

11. The function of fostering social responsibility and accountability via the enforcement of consequences for the conduct of institutions and people: This function of journalism is crucial in cultivating a culture of openness and ethical institutional conduct since it aids in uncovering any misconduct or malfeasance inside organizations and people. Journalists have a crucial role in upholding social responsibility and ensuring accountability by bringing attention to these concerns. In essence, it fosters a society characterized by fairness and honesty, ensuring equal treatment for all individuals.

12. The function of questioning established norms and promoting analytical thinking means that journalists, in addition to motivating action to solve widespread problems and implementing innovative responses both in institutional and community terms, promote in people the development of their analytical capacity in problematic situations, as well as the questioning of the norms established to promote progress. This can lead to a more informed and critical society, where individuals are encouraged to question authority and challenge the status quo. By promoting analytical thinking, journalists play a crucial role in fostering a culture of accountability and driving positive change in both institutions and communities.

## 4. Discussion and Conclusions

The complexity of today's societies, mired and often stuck in recurring crises, puts pressure on journalism to expand its limits. Therefore, it is a professional field that is constantly changing. The important normative question is frequently asked about what the ideals and practices of good journalism should be. If the explanatory and constructive functions of journalism are considered desirable, it is on the basis that these professional models have the social responsibility of improving society. In order to fulfill these functions effectively, journalism must adapt to the evolving needs and expectations of its audience. This requires journalists to not only report on events but also provide context, analysis, and solutions to the complex issues faced by society. By embracing innovation and new technologies, journalism can continue to play a crucial role in shaping public discourse and fostering positive change.

This article has fulfilled its main objective of exploring and analyzing the role of journalists in the current circumstances of crisis in democratic societies and assessing constructive models and solutions for journalistic work from this perspective. It has been possible to formulate a set of constructive journalistic roles, which generally present a more comprehensive and balanced perspective on important problems of society. There is a growing emphasis on solutions journalism, which focuses on reporting not just on problems but also on potential solutions and positive change. By adopting these roles, journalists can contribute to fostering a more informed and engaged citizenry, ultimately strengthening democratic societies.

Some of the journalistic roles proposed by Krüger et al. (2022), Hermida and Mellado (2020), Standaert et al. (2019), McIntyre et al. (2018), Abdenour et al. (2018, 2021), Dahmen et al. (2019), Li (2023), Thier and Namkoong (2023), Mäder and Rinsdorf (2023), Hanitzsch and Vos (2018, 2017), Ojala and Pöyhtäri (2018), Hanitzsch et al. (2016), Tandoc and Peters (2015), Mellado and Dalen (2014), and Dalen et al. (2012) in their investigations have been taken and expanded, and particularly, they have been reoriented towards the explanatory and constructive nature of the journalism model that has been exposed in this work. Even some journalistic functions, such as contextualization, promotion of public interest, dialogue, and understanding, as well as motivation or inspiration to action and commitment to communities and societies, are considered fundamental beyond constructive journalism or solutions.

This approach highlights the importance of contextual journalism and the influence of the roles and values of journalists in their coverage of various topics, with the public highly valuing the roles of contextual journalism, following the contributions of McIntyre et al. (2018), Abdenour et al. (2018, 2021), Dahmen et al. (2019), Li (2023), Thier and Namkoong (2023), and Mäder and Rinsdorf (2023). The roles associated with contextual

reporting place high value on being socially responsible and accurately portraying the world. Journalists' belief in activist values, such as setting the political agenda and pointing at possible solutions, predicted more favorable views of all three forms of contextual journalism. The most prized function was the hybrid, which combines interpretive roles with 'just the facts' disseminator roles. Job function heavily influences attitudes toward this reporting style, with the advocate/entertainer function being the strongest overall predictor of positive attitudes toward contextual journalism genres.

These roles of journalists are based on a general claim that progressivism is of major importance to society. The claim that progressivism is good for society is not universally accepted, as there are differing opinions on the effectiveness and impact of progressive policies. However, journalists who adopt these roles believe that by shedding light on hidden truths and promoting positive change, they can help create a more equitable and just society. It is important for journalists to remain objective and provide balanced coverage to ensure that their work resonates with a wide and diverse range of readers and viewers.

Media outlets, both traditional and online, and experts in the fields of information and public opinion have all played a role in either establishing or reinforcing a pervasive and depressing feeling of hostility and pessimism, especially through their coverage of difficult situations and crises that fail to provide adequate context and instead magnifies the severity of the problems. Particularly relevant is the fact that they have exerted effort to establish or enhance this perception (Sacerdote et al. 2020; Aslam et al. 2020). Society has suffered greatly as a result of this negativity and strife, which encourages a gloomy view and impedes the pursuit of positive alternatives. Professionals and media organizations should work toward fair reporting that helps the public comprehend complicated issues from all angles. This will lead to a more sophisticated and educated discussion on these topics.

It is evident that the media play a vital role in shaping both public perception and trust in democratic processes and public authorities. To restore this faith, it is essential to focus on hopeful narratives, solutions-oriented coverage, and actions that tackle issues of public concern. By adopting these strategies, the media can play a constructive role in fostering a more positive and engaged society, ultimately leading to a stronger and more resilient democracy (Rojas-Calderón 2023a). By providing balanced and solution-oriented coverage, media outlets can help foster a more informed and engaged citizenry, leading to more constructive political discourse and decision-making. This, in turn, may contribute to greater public trust in democratic institutions and processes, as well as a stronger sense of civic responsibility among individuals.

In this way, constructive journalism formats can serve as a powerful tool to counter ideological and discursive extremism among political actors, even in an environment that favors individual and direct expression. By presenting balanced and solutions-oriented coverage, constructive journalism can provide a counter-narrative to extreme viewpoints, promoting a more nuanced and informed understanding of complex issues. This can help to reduce polarization and encourage constructive dialogue, ultimately fostering a more constructive political environment. Furthermore, by engaging with diverse voices and perspectives, constructive journalism can help to foster a sense of inclusivity and mutual understanding, thereby undermining the appeal of divisive and exclusionary rhetoric.

By embracing this new explanatory and constructive model, journalists can actively address the existing biases in news reporting and work towards a more inclusive and balanced representation of diverse perspectives. Additionally, it allows for a deeper understanding of the societal factors that influence news content, enabling journalists to present information in a manner that fosters meaningful dialogue and encourages positive actions within communities. Furthermore, this approach encourages journalists to engage with their audience and seek feedback, creating a collaborative relationship that promotes transparency and accountability. By actively involving the public in the news-making process, journalists can ensure that their reporting is relevant and impactful, ultimately leading to a more informed and empowered society.

However, as advice for journalistic practice, as indicated by Meier (2018), a considered use of constructive journalism could be integrated into newsroom strategies. First, the hopeful prospects should not be used to simply garnish a difficult problem at any price, and secondly not be at the expense of a differentiated and comprehensive presentation of a complex social problem. In consonance with (Hermans and Drok 2018), journalism needs to move in a new direction, fostering cooperation, transparency, and constructiveness. In this order, The key innovation for the coming years will be to put citizens, in their capacity as potential actors, in the center of journalism. At stake is the more fundamental question about the function of professional journalism in the network era on all levels: individual, community, and society. Journalism culture, its goals, values, and norms must be attuned to the changing circumstances to prevent journalism from widening the gap between citizens and society.

This work, considering its preliminary and exploratory nature, had the limitation that it did not have an empirical base. Indeed, this should be the purpose of future research on explanatory and constructive journalistic roles. People worry and demand more information about social issues while getting torn by being drawn into negativity and conflict. This shift toward explanatory and constructive reporting allows journalists to address these concerns by providing a balanced perspective that goes beyond sensationalism. The synthesis of common categories has provided a comprehensive understanding of the various journalistic roles within the context of this investigation. This will enable researchers and practitioners to bridge the gap between theoretical knowledge and practical implementation. However, further research and validation are required to ensure the accuracy and effectiveness of these newly developed categories in real-world scenarios.

This progressive approach to journalism can help close the gap between information and action, giving readers a deeper understanding of complex social problems and their potential solutions. By focusing on solutions and highlighting the efforts of people and organizations working for positive change, journalists can inspire readers to take action and engage in constructive dialogue. This approach not only fosters a more informed society but also cultivates a sense of hope and agency among community members.

According to From and Kristensen (2018), constructive journalism may revitalize journalism's role in society by taking on a solution-oriented and positively inclined service role in both public and private life matters of audiences. However, critics argue that applying such a normative and positive approach could potentially neglect grim aspects of reality and distance journalism from key values such as objectivity, autonomy, and a critical approach, potentially losing credibility and appearing less authoritative. In this sense, constructive journalism and service journalism share a commonality in their approach and mode of address.

**Funding:** This research received no external funding.

**Institutional Review Board Statement:** Not applicable.

**Informed Consent Statement:** Not applicable.

**Data Availability Statement:** No new data were created.

**Conflicts of Interest:** The authors declare no conflict of interest.

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
