# Peer review of "Constructive Journalistic Roles in Environments of Social Complexity and Political Crisis"

_journalmedia, doi:10.3390/journalmedia5020042_

Round 1
Reviewer 1 Report
Comments and Suggestions for Authors
I think this study sets out with a good and justifiable purpose to help understand and synthesize much of the journalistic role discourse in current research. There are a lot of studies with affirming and contradicting findings, and plenty of opportunity for an essay like this.
However, I find the manuscript as it stands to lack the necessary critique, analysis and synthesis necessary in order to accept the argument and proposed formulated roles. The author(s) set forth a detailed list but it's hard to understand why and how these terms should be accepted and used as a foundation for further study. This is not saying I think this should be an empirical study, I agree with the author(s) saying it is preliminary and exploratory.
There are many terms discussed in the paper, and there needs to be a stronger definition and distinction about each of them. Solutions and constructive journalism are different approaches, though heavily related (many position solutions under the broader constructive umbrella). Explanatory, contextual, etc, terms are mentioned often without proper connection or explication to understand the thread the author(s) are weaving. I see where the author(s) are going with this essay, but feel the narrative gets lost amidst terms sometimes.
Additionally, while there is a comprehensive and thorough list of journalistic role studies discussed in the paper, they are mainly just summarized without much synthesis or critique in a productive manner that moves us toward the conclusions. I appreciate the depth of writing on the studies, but need to see more analysis and critique that gets beyond simple description of the studies.
I think this weakens the overall intended result of the essay because it makes it hard to understand the key anchors for the 12 roles proposed by the author(s). In order to agree that this list is superior to the prior studies, and adds something rigorous and new to explore, I think the author(s) need to spend more time synthesizing and justifying how they arrived at these conclusions. Otherwise it is hard to be able to find credibility in the argument.
A few minor notes:
- On page 2, paragraph that begins on line 73, I think the author(s) are correctly explaining how constructive reporting IS critical, independent, etc and fulfills these roles in ways that shallow "positive" or "good news" reporting may not, but this could be more clearly stated.
- On P5, the two paragraphs devoted to a study on metajournalistic discourse. It is important to notice that the majority of that dataset came from voices inside the respective organizations and thus the findings should be interpreted with caution since it is built primarily on internal discourse. In many ways it simply reflects the internal definitions, etc, without hearing from external, independent voices who are involved, such as journalists who have practiced it, critics, etc.
- The contextualist study on p7 line 323 has done a lot of the synthesis in an empirical way, and the authors tested it in several other studies that may be of use to this paper. Key to this is how their work involves input from both journalists and the public. I'd recommend looking at the full set of studies and seeing how their findings might contribute here.
Comments on the Quality of English LanguageReadability is good, just some awkward grammar/phrasing in places that will be smoothed out during copy editing.
Author Response
Thank you for your observations and recommendations.
- More criticism, analysis and synthesis has been introduced, both of the theoretical contributions and the proposal of journalistic roles.
- Although this is a preliminary and exploratory work, the foundations and scope of the role proposal have been better explained.
- A clarification has been introduced on the use of the concepts of constructive journalism and solutions journalism. Indeed, we agree that the constructive approach may be a little broader due to its development, but we join the authors who use these terms interchangeably.
- The narrative was reviewed and improved to avoid confusion with explanatory, contextual and constructive terms, among others.
- More analysis, criticism and relationship between the concepts and categories of the cited studies have been introduced. This constitutes the foundation of the roles that have been formulated.
- Contributions from other works and authors have been incorporated to expand the theoretical support of the proposal to summarize journalistic roles.
- The confusions indicated on pages 2 (on superficial positive or good news), 5 (on the limitations of studies on metajournalistic discourse) and 7 (other findings on contextualist studies) have been corrected and clarified.
Thanks
Reviewer 2 Report
Comments and Suggestions for Authors
I commend the author(s) on tackling this topic - it is one worth exploring. But in its present form, I do not think this manuscript adequately achieves its goal of "analysis of secondary sources to propose professional roles for journalists to revitalize their roles in social and political coexistence. It aims to bridge the gap between theoretical concepts and practical applications." To begin, throughout the paper are multiple examples of the author(s)' opinion when the statements should be cited. (Here are two of several examples: "By embracing these various functions, journalists are able to provide a more comprehensive and balanced perspective on important issues, fostering a more informed and engaged society. This shift towards a constructive and positive approach not only enhances the credibility of journalism but also strengthens democracy by empowering individuals to actively participate in shaping their communities." and "This means that journalists do not only report on events and provide information but also analyze and interpret the significance of these events for the public itself. They go beyond just presenting facts and strive to uncover the underlying truths and implications of various situations. This in-depth analysis helps citizens make informed decisions and encourages critical thinking, ultimately leading to a more democratic society where people are actively engaged in those issues affecting them."). Second, the article needs more text that explains to the reader the order/your approach to present your evidence and then conclusions to help guide the reader to see how you arrived at your conclusions. For instance, at line 296 it seems that the manuscript switches to role conceptions about constructive/solutions journalism, but this isn't clear to the reader. Furthermore, I would suggest integrating the Proposition section with your review of constructive/solutions literature, perhaps with one section for each proposition. As it stands, it is hard to see how you move from the literature to the development of the propositions. Additionally, the title suggests this manuscript is about explanatory and constructive journalism, but the review seems to be about constructive. I would stick with one and amend the title. Similarly, the title suggests that these new journalistic roles are required due to “social complexity and political crisis”; however, these elements nor how journalism/explanatory and constructive journalism relate to them are discussed.
Finally, while you examine much of the literature about constructive/solutions journalistic role conceptions, there are several key studies left out. First, Thier & Namkoong address role conceptions in their review, "Identifying Major Components of Solutions-Oriented Journalism: A Review to Guide Future Research." Not only do they offer conclusions that may be relevant to your work, but you will find some additional articles for your study within their review. Other studies that come to mine are Li (2023), "Assessing the role performance of solutions journalism in a global pandemic" and Mader & Rinsdorf (2022), "Constructive Journalism as an Adaptation to a Changing Media Environment." Constructive/solutions scholar Kyser Lough maintains a public bibliography of such literature and from a quick look at it, there seem to be other studies which should be included/reviewed in yours. Reference below. http://www.kyserlough.com/solutionsjournalism.html
Mäder, A., & Rinsdorf, L. (2023). Constructive Journalism as an Adaptation to a Changing Media Environment. Journalism Studies, 24(3), 329-346.
Thier, K., & Namkoong, K. (2023). Identifying Major Components of Solutions-Oriented Journalism: A Review to Guide Future Research. Journalism Studies, 24(12), 1557-1574.
Li, Y. (2023). Assessing the role performance of solutions journalism in a global pandemic. Journalism Practice, 17(7), 1445-1464.
Comments on the Quality of English LanguageMinor revisions needed.
Author Response
Dear Dr. Maksimovic
I´m sending you the revise versión of manuscript.
I know I´m late. I´m so sorry.
Kind regards.

Round 2
Reviewer 1 Report
Comments and Suggestions for Authors
I think the manuscript has been greatly strengthened and is clearer in both its purpose and synthesis.
Author Response
Thanks for your review. Your observations and contributions have been very important, so that the manuscript has improved and is more complete.
I hope you can sign your review report.

Reviewer 2 Report
Comments and Suggestions for Authors
I appreciate the work this authors did in incorporating the studies I suggested. I think this has strengthened the manuscript. (One note, Thier & Namkoong (2023) is misspelled several times throughout the manuscript.) However, I still feel the manuscript does not adequately explain how the authors moved from reviewing the literature to developing their propositions. What approach was followed? Without this it is unclear whether the propositions are appropriate and whether the article is contributing something new to the literature.
Comments on the Quality of English LanguageThere are some places where the subject-verb agreement and other grammar issues should be addressed. However, I think the overall quality is high.
Author Response
We are sending a new version of the manuscript with the latest corrections that you have suggested. We revised the writing and expanded the transition from reviewing literature to developing propositions. Your observations and contributions have been very important in making the manuscript better and more complete.
We hope you can sign your review report.
